# Tensor-Free Holographic Metasurface Leaky-Wave Multi-Beam Antennas with Tailorable Gain and Polarization

**DOI:** 10.3390/s24082422

**Published:** 2024-04-10

**Authors:** Chuan-Kuei Weng, Yu-Zhan Tsai, Artem Vilenskiy, Malcolm Ng Mou Kehn

**Affiliations:** 1Institute of Communications Engineering, National Yang Ming Chiao Tung University, Hsinchu 300093, Taiwan; panzer8792.ee10@nycu.edu.tw (Y.-Z.T.); malcolm.ng@ieee.org (M.N.M.K.); 2Department of Electrical Engineering, Chalmers University of Technology, SE 41296 Goteborg, Sweden; artem.vilenskiy@chalmers.se

**Keywords:** holographic antennas, leaky-wave antennas, metasurfaces, millimeter wave antennas, multiple beams

## Abstract

Recently, the community has seen a rise in interest and development regarding holographic antennas. The planar hologram is made of subwavelength metal patches printed on a grounded dielectric board, constituting flat metasurfaces. When a known reference wave is launched, the hologram produces a pencil beam towards a prescribed direction. Most earlier works on such antennas have considered only a single beam. For the few later ones that studied multiple beams, they were achieved either by having each beam taken care of by a distinct frequency or by partitioning the hologram, thereby depriving each beam of the directivity it could have had it not shared the holographic aperture with other beams. There have been recent studies related to the use of tensor surface impedance concepts for the synthesis of holograms which have attained control over the polarizations and intensities of the beams. However, this approach is complicated, tedious, and time-consuming. In this paper, we present a method for designing a planar holographic leaky-wave multi-beam metasurface antenna, of which each simultaneous beam radiating at the same frequency towards any designated direction has a tailorable amplitude, phase, and polarization, all without hologram partitioning. Most importantly, this antenna is exempted from the need for the cumbersome technique of tensor impedance. Such features of beam configurability are useful in selective multiple-target applications that require differential gain and polarization control among the various beams. Only a single source is needed, which is another benefit. In addition, effective methods to mitigate sidelobes are also proposed here. Designs by simulations according to the method are herein validated with measurements performed on fabricated prototypes.

## 1. Introduction

Common high-gain antennas, such as reflector antennas, lens antennas, and phased array antennas, face drawbacks due to their complexity, heavy weight, and high cost. In recent years, the design of holographic antennas has attracted considerable attention due to their advantages of being small in size, lightweight, low-cost, and simple configuration as well as the potential to achieve high gain, circular polarization, and beamforming.

Holographic antennas are a kind of directional leaky wave antenna (LWA) which can be designed based on the theory of holography [1,2,3]. The holographic plane consists of a grounded printed circuit board (PCB) and sub-wavelength conducting patches imprinted onto the surface. When excited by a known reference surface wave, the holographic plane produces a pencil beam towards a designated direction. Through rearrangement of the metal patches, different radiation angles can be configured.

Originating from the field of optics, holography is a technology for recording an object’s optical information onto a plane called a hologram and the subsequent reproduction of its image when light passes through the plate, also known as an interferogram. This idea can also be extended to the microwave arena [4,5], such as applied to the design of LWAs in millimeter wave bands. During earlier times, there were two common types of holographic antennas [6], namely reflected type and transmitted types, although both of those suffer from large scattered wave losses. In recent years, researchers have begun to develop newer holographic antennas, the source and interferogram of which are on the same plane, thereby alleviating transmission losses and also reducing the volume of the entire system. This concept is illustrated in Figure 1a for an example of a semicircular hologram adequate for forward-directed beams, which depicts multiple beams simultaneously radiated from a single flat holographic LWA due to surface waves propagating along the artificial impedance surface (AIS) synthesized according to the interference patterns of various object waves bearing information of beam directions with known reference waves. Figure 1b schematizes the image reconstruction of an object comprising distributed sources achieved by a generic form of circularly shaped planar holographic LWA made of metallic patches printed on a grounded substrate.

In the next generation of 6G wireless communications, there will be skyrocketing demands for higher data speeds, enhanced accuracies, bandwidths, spectral efficiencies, energy efficiencies (low energy and power consumption), higher throughputs and transmission capacities, low latencies, massive connectivity, higher connection densities, and wider coverages. Apart from communications, the vast range of applications of 6G technologies include radar, tracking, imaging/spectrometry (e.g., medical diagnoses), detection (such as for safety and security against, e.g., concealed weapons, explosives, illicit substances, contraband goods, etc.), scanning, inspection (for mechanical damage, food contaminants, e.g., in food research), facial and gesture recognitions, automotive applications (adaptive cruise control, blind spot detection, lane change and parking assistance, collision avoidance, etc.), autonomous systems, entertainment, Internet of things (IoT), smart homes and cities, positioning, localization, and sensing (e.g., environmental sensing, material sensing) amongst many others. The latter features are of ever-increasing importance as they are among the cutting-edge, futuristic paradigms of 6G systems. Enhancements in localization would be the pursuit of more accurate outdoor positioning beyond what global positioning systems (GPS) are able to provide, even up to centimeter-level accuracies, accurate indoor positioning that calls for interpretations of the third dimension in addition to the two-dimensional information afforded by existing technologies, and incorporation of three-dimensionality into localization via discernment of roll, pitch, and yaw [7].

There have been studies on localization methods, such as phase- and angle-based positioning, 3D millimeter wave tracing, as well as map-based and map-assisted localizations [8]. Other new techniques recently conceived include the use of reconfigurable intelligent surfaces (RIS) or intelligent reflective surfaces (IRS) [9] as well as MIMO (multiple-input, multiple-output) arrays, whereby the fullness of the localized object is enhanced by the multiple antennas. In massive versions, i.e., massive MIMOs (mMIMOs), the large numbers of antennas in the transmitters and receivers have led to the establishment of multiple signal propagation paths needed for localization and sensing. The multiple beams thus have to be highly directive and reconfigurable/steerable in order to attain high resolution spatial multiplexing [10]. Also known as holographic radio or holographic MIMO (HMIMO), mMIMO is a revolutionary technology that can potentially redefine the boundaries of 6G localization and sensing. It also connects to the important feature of beamforming, which controls the mobile radio channel or tailors the wireless propagation environment with reconfigurable (programmable) intelligent surfaces, typically using metasurfaces.

Holographic localization is a technology whereby the properties or propagation regimes (including the near-field regions) of electromagnetic (EM) waves transmitted or received by antennas of multiple devices are optimally manipulated by associating with the phase profile of the impinging spherical wave-front curvature in the near field. This provides more information about the location and orientation of the target, thereby achieving improved wireless localization [11]. With HMIMOs or mMIMOs entailing extremely large (in terms of wavelength) antenna arrays, the high spatial resolution and wide bandwidths that they afford pave the way towards rapid and accurate wireless positioning. Their near-field regimes are also larger, rendering users more likely to be located within the near-field zones, thereby offering more degrees of freedom that facilitate the retrieval from the spherical wave-front, knowledge of the distance (range) and direction (angle of arrival) of the localized object. In other words, signals can be pinpointed at precise spots (information of both the distance and direction), which constitutes a major advancement from just knowing the direction. Not only does this boost the received signal-to-noise ratio (SNR) at the user, interferences with other users are also mitigated, all of which leading to accurate estimations of the users’ locations.

A majority of earlier reports on holographic antennas has considered only a single beam, e.g., [12,13]. Among the minority which studied multiple beams, individual beams were achieved either by radiating each one at a distinct frequency [14,15] or by partitioning the hologram [16], thereby depriving each beam of the directivity it could have had it not shared the holographic aperture with other beams. Prior studies have reported the use of tensor surface impedance elements for achieving control of polarization and phase of the radiation, e.g., [17,18,19,20]. Moreover, most reported modulated metasurfaces only produced a limited number of beams, although a study of five beams emitted by a superposed holographic pattern was conducted in [21]. Despite the study in [22] of multiple beams achievable by a superposed holographic pattern, it was however only for radiation of a single linear polarization within a single vertical (azimuth) plane by a linear array. In [23], other than also being limited to the treatment of beams confined to a single plane and a single polarization, neither differential amplitude control of beams nor experimental work was reported in that work. It also entailed the use of varactor diodes with associated drawbacks to tailor the required surface properties. There is also no amplitude or polarization control in [24] despite the beam multiplicity studied there.

In spite of the abovementioned works that have engaged the method of tensor surface impedances for the synthesis of holographic LWAs with beam multiplicity—each of a prescribed direction, amplitude, and polarization—that approach is complicated, time consuming, and cumbersome. Contrarily, in this paper, we propose a far simpler and less tiresome method under typical practical reference wave scenarios requiring only simple scalar surface impedances to design holographic LWAs entailing two-dimensional planar arrays of periodic metal patches on grounded substrates known as metasurfaces that can achieve simultaneous multiple-beam radiation all at the same frequency without hologram partitioning as well as independent amplitude, phase, and polarization control of each beam that can be within any azimuth plane. The latter feature is useful in selective multiple target applications that require differential gain apportionment and polarization distinction among the various beams towards any direction not limited to a single plane. Despite the multiplicity of beam angles, just a single source is required for our method, constituting a further benefit. In addition to different linear polarizations in multiple directions, our design can also be configured for two orthogonally polarized beams with a 90° phase difference between them that radiate towards the same direction to achieve circular polarization. All these capabilities of generating multiple directive beams of tailorable direction, amplitude, phase, and polarization enable our herein-reported holographic LWA to be highly suitable for use in 6G localization and sensing applications. A millimeter wave frequency of 28 GHz is herein considered as opposed to other common bands lower down the spectrum. A preliminary version of our work was reported in [25], which is herein updated and expanded.

Capitalizing on the amplitude, phase and polarization control afforded by our approach, circular polarized radiation as well as a method for the suppression of sidelobes of the holographic LWA by beam cancellation are showcased. For the latter, by superposing holograms which radiate beams that are in anti-phase with the targeted sidelobe, its level can be effectively reduced. Although sidelobe suppression by beam cancellation is not new and similar studies have been reported [26], all of them suffered from weakened main beams, which may be the result of inadequate rigor in their designs rather than an inevitable tradeoff. In our approach, in addition to the sidelobe-cancelling hologram that masks over the original one responsible for the main beam, another compensatory one to make back up for the compromised main beam is further superposed, thereby achieving sidelobe mitigation as well as main beam enhancement to levels even beyond the original one, as shall be demonstrated in this work.

The formulation from first principles of the tensor surface impedance theorem towards the simplification to just scalar impedances for the syntheses of holograms excited by common realizable reference beam sources is first presented in Section 2. Prior to the design of the unit cell in Section 4, the design study of dual-beam radiation with dual-polarization by a holographic LWA using tensor impedance is presented in Section 3, which serves as a background for the investigations in Section 5 of simultaneous multiple beams with independent amplitude and polarization controls. The use of the approach to achieve circular-polarized radiation by the holographic LWA is presented in Section 6 and the method of sidelobe suppression by beam cancellation is described in Section 7. A prototype with dual-beam capability based on the numerical design was manufactured, and the measured results in terms of the matching and far-field radiation patterns are presented in Section 8. After comparing our work to that of literary contemporaries in Section 9 and discussing the differences of holographic antennas from phased arrays and sector antennas in Section 10, the paper is concluded with a summary in Section 11.

## 2. Simplification of the Tensor Impedance

According to [1], already knowing that tensor impedances can achieve polarization control, we still by that same cue follow out the upcoming formulations from the fundamentals of tensor surface impedances. Then, upon the simplification afforded by our practical and commonly realizable reference wave, we branch off towards the reduction to the entailment of just simple scalar surface impedances for the synthesis of our holographic metasurface, LWA, thus circumventing the unwieldy tensors and verifying our much simpler and more convenient technique.

### 2.1. Surface Impedance Tensor

With *F* denoting either *E* or *H* for representing the surface components of the fields, each field vector is expressed as a sum of the two modal groups by
(1)$$F^{TOT}={\left[\begin{array}{cc} F_{\rho }^{TOT} & F_{\phi }^{TOT} \end{array}\right]}^{Τ}=F^{TM}+\alpha F^{TE}$$
where the superscript T signifies non-conjugate transposition and *α* is the arbitrary ratio of the TE modal amplitude to that of the TM mode. Furthermore, with $\;\hat{x}$ and $\;\hat{y}$ being the unit vectors along *x* and *y*, with corresponding wavenumbers or phase constants *k_x_* and *k_y_* along those respective directions, then
(2)$${\;\hat{\rho }}_{surf}=\left(\;\hat{x}k_{x}+\;\hat{y}k_{y}\right)/\sqrt{k_{x}^{2}+k_{y}^{2}}$$
is the radial unit vector along the direction of surface wave propagation with wavenumber
(3)$$k_{surf}=\sqrt{k_{x}^{2}+k_{y}^{2}}$$
with associated
(4)$${\hat{\phi }}_{surf}=\;\hat{z}\times {\hat{\rho }}_{surf}$$
being the unit vector along the azimuth *ϕ* angular coordinate, in which $\;\hat{z}$ is the unit vector along *z*, and
(5)$${\hat{\rho }}_{obs}=\left(\;\hat{x}x_{obs}+\;\hat{y}y_{obs}\right)/\sqrt{x_{obs}^{2}+y_{obs}^{2}}$$
is the radial unit vector towards an observation point (*x_obs_*, *y_obs_*) at a radial distance of
(6)$$\rho_{obs}=\sqrt{x_{obs}^{2}+y_{obs}^{2}}$$
from the origin. Then, with
(7)$$\Omega =\left\{\begin{array}{c} E\\ H \end{array}\right\},\;℧=\left\{\begin{array}{c} H\\ E \end{array}\right\},\;\zeta =\left\{\begin{array}{c} TE\\ TM \end{array}\right\},\;V=\left\{\begin{array}{c} Z_{0}\\ 1 \end{array}\right\},\;\Lambda =\left\{\begin{array}{c} 1\\ Z_{0} \end{array}\right\}$$
in which the two items within the curly braces in any given equation correspond to one another throughout that relation, and where *Z*_0_ = 1/√(*μ*_0_*ε*_0_) is the free-space impedance, the TM and TE modal fields are written succinctly as
(8)$$\Omega^{\zeta }=V{\hat{\phi }}_{surf}k_{surf}\Psi$$
(9)$${℧}^{\zeta }=\Lambda \frac{1}{k_{0}}\left(\left\{\pm \right\}\;\hat{z}k_{surf}^{2}+\left\{\pm \right\}j\alpha_{z}{\;\hat{\rho }}_{surf}k_{surf}\right)\Psi$$
(10)$$\Psi =\exp\left(-j{\;\hat{\rho }}_{surf}k_{surf}\cdot {\;\hat{\rho }}_{obs}\rho_{obs}\right)\exp\left(-\alpha_{z}z\right)$$
whereby *α_z_* is, for surface waves, the attenuation constant along *z* perpendicular to the surface and *k*_0_ = *ω*√(*μ*_0_*ε*_0_) is the free-space wavenumber, with *ω* = 2*πf* being the angular frequency.

The surface electric field and current are related via the tensor surface impedance $\;\bar{Z}$ according to
(11)$$E^{TOT}=\;\bar{Z}\cdot J$$
where
(12)$$\;\bar{Z}=\left[\begin{array}{cc} Z_{\rho \rho } & Z_{\rho \phi }\\ Z_{\phi \rho } & Z_{\phi \phi } \end{array}\right]$$
(13)$$J={\left[\begin{array}{cc} J_{\rho } & J_{\phi } \end{array}\right]}^{Τ}=\;\hat{z}\times H^{TOT}$$

Equating ${\;\hat{\rho }}_{surf}$ and ${\hat{\phi }}_{surf}$ components on both sides of this latter and then combining the two relations, the following is derived.
(14)$$\frac{\alpha_{z}}{k_{0}}=\frac{\left\{j\left(Z_{\rho \phi }Z_{\phi \rho }-Z_{\rho \rho }Z_{\phi \phi }-Z_{0}^{2}\right)\pm j\sqrt{{\left(Z_{\rho \phi }Z_{\phi \rho }-Z_{\rho \rho }Z_{\phi \phi }-Z_{0}^{2}\right)}^{2}-4Z_{\rho \rho }Z_{\phi \phi }Z_{0}^{2}}\right\}}{2Z_{\phi \phi }Z_{0}}$$

### 2.2. Holography Theorem

For an object wave of *E*-field $E_{obj}={\left[\begin{array}{cc} E_{\rho }^{obj} & E_{\phi }^{obj} \end{array}\right]}^{Τ}$ and a reference wave of *H*-field $H_{ref}={\left[\begin{array}{cc} H_{\rho }^{ref} & H_{\phi }^{ref} \end{array}\right]}^{Τ}$ which manifests a surface electric current
(15)$$J_{ref}=\;\hat{z}\times H_{ref}={\left[\begin{array}{cc} -H_{\phi }^{ref} & H_{\rho }^{ref} \end{array}\right]}^{Τ}={\left[\begin{array}{cc} J_{\rho }^{ref} & J_{\phi }^{ref} \end{array}\right]}^{Τ}$$
the interference pattern $\;\bar{\Pi }$ is written as
(16)$$\;\bar{\Pi }=E_{obj}⊗J_{ref}^{†}$$
in which ⊗ symbolizes the tensor or outer product and the † superscript denotes the Hermitian conjugate (conjugated transposition). The form of this (16) is such that
(17)$$\;\bar{\Pi }\cdot J_{ref}={\left|J_{ref}\right|}^{2}E_{obj}$$
thus fully recovering the object *E*-field.

Comparing this (17) with (11), it is readily recognized that the interference pattern $\bar{\Pi }$ matrix bears the same significance as the impedance tensor matrix $\;\bar{Z}$.

For the present TM reference cylindrical wave readily producible in practice by common vertical monopole sources, with only a single nonvanishing $H_{\phi }^{ref}$ component (particularly, $H_{\rho }^{ref}$ = 0), the simplified reference electric current vector of (15) results in a likewise simplified interference pattern in place of (16), expressed as
(18)$$\;\bar{\Pi }=-\left[\begin{array}{cc} E_{\rho }^{obj}H_{\phi }^{ref*} & 0\\ E_{\phi }^{obj}H_{\phi }^{ref*} & 0 \end{array}\right]$$

It is then this latter interference pattern, representative of a tensor impedance matrix as asserted earlier, that is to be synthesized using metasurface elements into a holographic impedance surface distribution, expressed as
(19) Z¯surfholo(ρ,ϕ)=j[X00X]{1+MIm( Π¯)}=aswrite[ZρρZρϕZϕρZϕϕ]where *X* is the average surface reactance of the span achievable by the range of parameters which the metasurface unit cell accommodates and *M* is the modulation depth, being the division of the difference between the maximum reactance and *X* by the latter.

In (18), it can be readily observed that *Z_ρϕ_* of (19) is zero. Thus, (14) becomes
(20)$$Z_{\rho \rho }\underset{Z_{\rho \phi }=0}{=}\frac{j\alpha_{z}Z_{0}}{k_{0}}=Z_{surf}^{TM}$$
the right-hand side, being the well-known TM surface impedance $Z_{surf}^{TM}$. Therefore, amongst the four tensor impedance elements, just a single one of them, *Z_ρρ_*, remains involved, and it equals this aforementioned surface impedance. Hence, we have proven the entailment of only a solitary scalar impedance for the synthesis of the hologram despite starting off first with principles from the impedance tensor formulation generally involving four tensor elements.

## 3. Simultaneous TM and TE Beams

In order to resolve and reconstruct object waves of different intensities arriving from different directions, and now additionally with different polarizations, we need to consider the vector information of the fields of the object waves. This is performed first for just a single object wave of each modal type, as follows. With lower-case *e* and *h* subscripts denoting TE and TM modes, respectively, the combined $E_{obj}$ field of the two object waves, one being TE polarized (${\hat{\phi }}_{0e}$ directed) of amplitude $E_{\phi_{0e}}$ arriving from (*θ*_0*e*_, *ϕ*_0*e*_) and the other TM (${\hat{\theta }}_{0m}$ directed) of amplitude $E_{\theta_{0m}}$ arriving from (*θ*_0*m*_, *ϕ*_0*m*_), is expressed as a function of spatial coordinates (*x_s_*, *y_s_*) = (*ρ_s_*, *ϕ_s_*) of locations on the hologram surface, each point with its associated unit vectors (${\hat{\rho }}_{s}$, ${\hat{\phi }}_{s}$), according to
(21)$$E_{obj}=E_{\phi_{0e}}\psi_{0e}{\;\hat{\phi }}_{0e}-E_{\theta_{0m}}\psi_{0m}{\;\hat{\theta }}_{0m}={\;\hat{\rho }}_{s}E_{\rho_{s}}^{obj}+{\;\hat{\phi }}_{s}E_{\phi_{s}}^{obj}+\;\hat{z}E_{z}^{obj}$$
in which, with *χ* denoting either *e* or *m*,
(22)$$\psi_{0\chi }=\exp\left[-jk_{0}\sin\theta_{0\chi }\left(x_{s}\cos\phi_{0\chi }+y_{s}\sin\phi_{0\chi }\right)\right]$$
(23)$${\;\hat{\phi }}_{0e}=\sin\left(\phi_{s}-\phi_{0e}\right){\;\hat{\rho }}_{s}+\cos\left(\phi_{s}-\phi_{0e}\right){\;\hat{\phi }}_{s}$$
(24)$${\;\hat{\theta }}_{0m}={\;\hat{\rho }}_{0m}\cos\theta_{0m}-\;\hat{z}\sin\theta_{0m}$$
(25)$${\;\hat{\rho }}_{0m}={\;\hat{\rho }}_{s}\cos\left(\phi_{s}-\phi_{0m}\right)-{\;\hat{\phi }}_{s}\sin\left(\phi_{s}-\phi_{0m}\right)$$
(26)$$E_{\rho_{s}}^{obj}=E_{\phi_{0e}}\psi_{0e}\sin\left(\phi_{s}-\phi_{0e}\right)-E_{\theta_{0m}}\psi_{0m}\cos\theta_{0m}\cos\left(\phi_{s}-\phi_{0m}\right)$$
(27)$$E_{\phi_{s}}^{obj}=E_{\phi_{0e}}\psi_{0e}\cos\left(\phi_{s}-\phi_{0e}\right)+E_{\theta_{0m}}\psi_{0m}\cos\theta_{0m}\sin\left(\phi_{s}-\phi_{0m}\right)$$
(28)$$E_{z}^{obj}=E_{\theta_{0m}}\psi_{0m}\sin\theta_{0m}$$

With the sole ${\;\hat{\phi }}_{s}$-directed *H*-field of the reference wave being
(29)$$H_{\phi_{s}}^{ref}=H_{0}\exp\left(-jk_{0}n\rho_{s}\right)$$
where *H*_0_ is an arbitrary coefficient and *n* is the average effective refractive index of the span attainable by the parametric range of the considered metasurface unit cell, the solely entailed *Z_ρρ_* element in the tensor matrix of (19) is written as
(30)Zρρ=jX[1+MIm(EρsobjHϕsref∗)]
where
(31)$$\begin{array}{l} E_{\rho_{s}}^{obj}H_{\phi_{s}}^{ref*}=E_{\phi_{0e}}H_{0}^{*}\sin\left(\phi_{s}-\phi_{0e}\right)\psi_{0e}\exp\left(jk_{0}n\rho_{s}\right)\\ -E_{\theta_{0m}}H_{0}^{*}\cos\theta_{0m}\cos\left(\phi_{s}-\phi_{0m}\right)\psi_{0m}\exp\left(jk_{0}n\rho_{s}\right) \end{array}$$

With all amplitude coefficients assumed real for simplicity of expression, the scalar impedance of (30) becomes
(32)$$\begin{array}{l} Z_{\rho \rho }=Z_{surf}^{TM}=j\alpha_{z}Z_{0}/k_{0}\\ =jX\left(1+M\left\{\begin{array}{l} E_{\theta_{0m}}H_{0}^{*}\cos\theta_{0m}\cos\left(\phi_{s}-\phi_{0m}\right)\sin\langle k_{0}\left[n\rho_{s}-\sin\theta_{0m}\left(x\cos\phi_{0m}+y\sin\phi_{0m}\right)\right]\rangle \\ -E_{\phi_{0e}}H_{0}^{*}\sin\left(\phi_{s}-\phi_{0e}\right)\sin\langle k_{0}\left[n\rho_{s}-\sin\theta_{0e}\left(x\cos\phi_{0m}+y\sin\phi_{0e}\right)\right]\rangle \end{array}\right\}\right) \end{array}$$

When just one of the two beams, each being either of the two modal polarizations, is desired for reconstruction, it is a simple task to retain only one of the two sum terms within the curly brace of this (32).

## 4. Design of the Unit Cell

The holographic plane is composed of sub-wavelength metal square patches of various sizes printed on a grounded dielectric substrate. The wavelength *λ* at 28 GHz in free space is about 10.7 mm, and each unit cell should be less than *λ*/4 (2.67 mm). For convenience, a universal unit cell size of 2 mm is chosen. A substrate made of Rogers 5880 (dielectric constant *ε_r_* = 2.2, loss tangent = 0.0009 and thickness 1.57 mm) was chosen for the design due to its low loss tangent that mitigates losses in the millimeter wave band. The unit cell has a square shape with a period of *p* = 2 mm. The structure is back-shielded by a copper sheet, and a metal patch is printed on the surface, as shown in Figure 2, with *g* denoting the gap size between adjacent patches.

To confirm that TM surface wave modes can propagate on the holographic plane comprising such cells, the CST eigenmode-solver was used for simulating the dispersion diagrams, being graphs of the surface wavenumber *k_s_* versus frequency. Upon obtaining *k_s_*, the TM surface impedance *Z_patch_* of different unit cells is given by [27]:(33)$$Z_{patch}=jZ_{0}\alpha_{z}/k_{0}$$
(34)$$\alpha_{z}=\sqrt{k_{s}^{2}-k_{0}^{2}}$$
where *Z*_0_ = √(*μ*_0_/*ε*_0_) is the intrinsic free-space wave impedance (377 Ω) and *α_z_* represents, as in (18), the positive real-valued decay constant along the surface normal (*z*-direction) [24,28].

Since any one gap size leads to a distinct *k_s_* for any given frequency *f* (thus *k*_0_), each in the considered range of gap sizes, from 0.4 to 1.2 mm, pertains to a certain *α_z_* of (34) and thus *Z_patch_* of (33). This allows us to numerically compute Z*_patch_* versus the gap size, as shown in Figure 3a for *f* = 28 GHz, in which *Z_patch_* ranges from *j*280 to *j*214 Ω. Therefore, *X* in (32) is obtained as *X* = (280 + 214)/2 = 247. Due to the sinusoidally oscillating nature of *Z_surf_* in (32), we need a factor to expand *X* to a maximum value of 280 and a minimum of 214 so that the surface impedance of the unit cell can match the hologram. Hence, we set *M* = (280 − 247)/247 = 0.133. Again, due to different *k_s_* associated with different gap sizes, the variation of the refractive index *n* at *f* = 28 GHz with the gap size can be plotted as shown in Figure 3b.

With these latter two graphs, the required gap size of the unit cell at any location on the hologram can be determined.

To verify the proposed formulation, the holographic antenna shown in Figure 4 was synthesized for two object beams of different incidence angles and polarizations using the above formula. The beam of the TM polarized object plane wave arrives from (*θ*_0*m*_ = 20°, *ϕ*_0*m*_ = 45°), while that of the TE polarization is from (*θ*_0*e*_ = 50°, *ϕ*_0*e*_ = 135°), both arbitrarily chosen directions. Equal amplitudes of both beams are assumed, i.e., *A^TM^* = *A^TE^* = 0.5.

For a center frequency of 28 GHz, the 3D gain patterns of the far fields simulated by CST are presented in Figure 5. The absolute gain (regardless of polarization) is depicted by Figure 5a, in which beams towards those two directions are clearly produced. Extracting the co-polar theta component of the *E*-field for the TM mode, it is seen in Figure 5b that only a single beam towards its prescribed direction (*θ*_0*m*_ = 20°, *ϕ*_0*m*_ = 45°) is now portrayed, with the radiation of this component towards the other unintended direction (*θ*_0*e*_ = 50°, *ϕ*_0*e*_ = 135°) of the TE polarization remaining weak. Likewise but conversely, strong radiation of the co-polar phi component of the *E*-field towards just the designated direction (*θ*_0*e*_ = 50°, *ϕ*_0*e*_ = 135°) for the TE case is observed in Figure 5c. The cross-polar phi component of the TM radiation towards its desired beam direction (*θ*_0*m*_ = 20°, *ϕ*_0*m*_ = 45°) is seen from Figure 5c to be weak as well. Similarly, the cross-polar theta component of the TE case towards its intended (*θ*_0*e*_ = 50°, *ϕ*_0*e*_ = 135°) is shown in Figure 5b to be also suppressed, as required.

## 5. Amplitude Control of Differently Polarized Beams

To realize a multi-beam holographic LWA that radiates both TM and TE polarized waves simultaneously with a configurable amplitude for each beam; particularly, *N_TM_* TM beams with the *i*-th one arriving from (*θ*_0*m*,*i*_, *ϕ*_0*m*,*i*_) and *N_TE_* TE beams with the *i*-th one arriving from (*θ*_0*e*,*i*_, *ϕ*_0*e*,*i*_), we rewrite the Formula (32) as
(35)$$\begin{array}{l} Z_{surf}=jX\\ \times \left\{1+M\left[\sum_{i=1}^{N_{TM}}\left(\begin{array}{l} A_{i}^{TM}\cos\theta_{0m}\cos\left(\phi_{s}-\phi_{0m,i}\right)\\ \times \sin\left\{k_{0}\left[\begin{array}{l} n\rho_{s}-\sin\theta_{0m,i}\\ \times \left(x\cos\phi_{0m,i}+y\sin\phi_{0m,i}\right) \end{array}\right]\right\} \end{array}\right)+\sum_{i=1}^{N_{TE}}\left(\begin{array}{l} A_{i}^{TE}\sin\left(\phi_{s}-\phi_{0e,i}\right)\\ \times \sin\left\{k_{0}\left[\begin{array}{l} n\rho_{s}-\sin\theta_{0e,i}\\ \times \left(x\cos\phi_{0e,i}+y\sin\phi_{0e,i}\right) \end{array}\right]\right\} \end{array}\right)\right]\right\} \end{array}$$
where *A_i_^TM^* and *A_i_^TE^* are the amplitudes of the *i*th beams of their respective polarizations, subjected to the condition of $\sum_{i=1}^{N_{TM}}A_{i}^{TM}+\sum_{i=1}^{N_{TE}}A_{i}^{TE}=1$.

In order to verify the formula, we consider two dual-beam holographic LWAs, each radiating two differently polarized beams. One of them has the same amplitude for both beams, and the other is where one beam is twice the strength of the other. Both designs share the same pair of beam directions, set to be towards *θ*_0*m*_ = 15°, *ϕ*_0*m*_ = 45° for the TM beam and *θ*_0*e*_ = 45°, *ϕ*_0*e*_ = 135° for the TE beam. For both these dual-beam cases, *N_TM_* = *N_TE_* = 1, and note the dropping of the beam index *i* since there is only one beam per polarization. As before, the CST solver is used for the simulations.

The simulated 3D gain patterns of the synthesized dual-beam, dual-polarized holographic LWA with equal amplitudes (*A^TM^* = *A^TE^* = 0.5) are presented in Figure 6. As the 3D radiation pattern of the absolute gain in Figure 6a portrays, the two beams are radiated with similar strengths toward their prescribed directions. For the TM-polarized beam, strong co-polar radiation along (*θ*_0*m*_ = 15°, *ϕ*_0*m*_ = 45°) is observed in the antenna pattern of Figure 6b for the theta component of the radiated gain defined by the E-field, whereas—as given in Figure 6c—the co-polar phi component of the gain pattern for the TE case displays a main beam towards (*θ*_0*e*_ = 45°, *ϕ*_0*e*_ = 135°). All respectively, the cross-polar phi and theta components of the E-field gain patterns in Figure 6b,c show weak radiation towards the designated beam directions of (*θ*_0*m*_ = 15°, *ϕ*_0*m*_ = 45°) and (*θ*_0*e*_ = 45°, *ϕ*_0*e*_ = 135°). These aspects are more clearly depicted by the planar plots in Figure 7 of these 3D patterns in those two phi cuts. *θ*_0*m*_ = 15° of the gain pattern in the *ϕ*_0*m*_ = 45° plane has strong and weak co-polar theta and cross-polar phi components, respectively. Exhibited in Figure 7b for the *ϕ*_0*e*_ = 135° cut are strong co-polar phi but weak cross-polar theta components along *θ*_0*e*_ = 45°. As required by the equal prescribed amplitudes for both cases of polarizations, the co-polar components are demonstrated by Figure 7a,b to be each radiated with an intensity of about 18 dBi.

We proceed to the next case of two differently polarized beams, with *A^TM^* = 0.333, *A^TE^* = 0.667, the latter being double the former. The simulated gain patterns in the same fashion as Figure 6 and Figure 7 are presented respectively in Figure 8 and Figure 9. Particularly, the 3D patterns of the absolute gain, the theta, and phi gain components are conveyed by Figure 8a–c respectively, and the patterns in the two phi planes containing the intended beams of both polarizations are given in Figure 9a,b, in each of which the patterns of both the co- and cross-polar components are presented. Without describing the details again, the same as before about the co- and cross-polar components of both polarizations (TM and TE) being strongly and weakly radiated towards their designated beam directions can also be said here for this case. The theoretical difference of 20log_10_(2) = 6.02 dB between both beam levels is seen to be demonstrated by the amount which the maximum gain level of the phi component in Figure 9b, being about 20 dBi, is stronger than that of the theta component in Figure 9a, being about 14 dBi.

In the next group of cases, two quadruple-beam holographic LWAs are studied. All four beams of the first are TM-polarized and emitted towards directions defined by (*ϕ*_0*m*,1_ = 150°, *ϕ*_0*m*,2_ = 120°, *ϕ*_0*m*,3_ = 60°, *ϕ*_0*m*,4_ = 30°) and (*θ*_0*m*,1_ = *θ*_0*m*,2_ = *θ*_0*m*,3_ = *θ*_0*m*,4_ = 30°) and have amplitude coefficients (*A*_1_^TM^ = *C*_1_ = 0.2, *A*_2_^TM^ = *C*_2_ = 0.15, *A*_3_^TM^ = *C*_3_ = 0.4, *A*_4_^TM^ = *C*_4_ = 0.25). It is observed from this schematic that because *C*_3_ is larger than the others, the interference pattern about the corresponding *ϕ*_0*m*,3_ = 60° region of the holographic plate is also more prominent.

Presented in Figure 10 and Figure 11 are likewise the respective 3D and 2D gain patterns, with any one of the four traces in the latter being that of a certain beam-containing phi plane. With all beams sharing the same *θ*_0*m*,*i*_ = 30° angle, it is more readily observed that the level differences among the gains indeed come to terms with the respective differences in their amplitude coefficients. For instance, the 4 dB drop from the peak gain (of about 19 dBi) of the beam with tailored strength of *C*_3_ = 0.4 to that (15 dBi) of the beam with *C*_4_ = 0.25 checks well with 20log_10_(0.4/0.25) = 4.08. The radiation and total efficiencies are *ε_rad_* = –0.18 dB (96%) and *ε_tot_* = –0.486 dB (89.4%), respectively.

For the second quadruple-beam holographic LWA, two beams are TM-polarized, while the other two are TE. This time, all of them share the same amplitude, i.e., *A*_1_^TM^ = *A*_2_^TM^ = *A*_1_^TE^ = *A*_2_^TE^ = 0.25. The radiation directions are (*θ*_0*m*,1_ = 15°, *ϕ*_0*m*,1_ = 150°), (*θ*_0*m*,2_ = 5°, *ϕ*_0*m*,2_ = 60°), (*θ*_0*e*,1_ = 45°, *ϕ*_0*e*,1_ = 120°), and (*θ*_0*e*,2_ = 55°, *ϕ*_0*e*,2_ = 30°), respectively.

The 3D patterns are presented in Figure 12, of which Figure 12a–c convey the absolute gain, the theta, and phi gain components, respectively. Equal strengths of radiation towards all four beam directions are observed in the first subplot, as required. Once again, the co- and cross-polar components of both polarizations (TM and TE) are strongly and weakly radiated towards their designated beam directions, as Figure 12b,c show for the TM and TE beams, respectively.

Planar plots of the patterns in the various phi cuts containing the four beams are presented in Figure 13. The two TM beams in the 150° and 60° phi planes are portrayed by Figure 13a, while the pair of TE ones in the 120° and 30° cuts are shown in Figure 13b. The theta and phi gain components are annotated in the legend of each subplot. Observed in Figure 13a for the TM beams are the radiations of strong co-polar theta but weak cross-polar phi gain components towards the two theta directions—*θ*_0*m*,1_ = 15° and *θ*_0*m*,2_ = 5° of both beams—whereas Figure 13b displays strongly and weakly radiated co-polar phi and cross-polar theta components towards *θ*_0*e*,1_ = 45° and *θ*_0*e*,2_ = 55°, respectively. Similar peak gains of about 16 dBi of the co-polar components are observed in Figure 13a,b, as expected of the equal prescribed amplitudes. For this case, the efficiencies are *ε_rad_* = –0.174 dB (96%) and *ε_tot_* = –0.4 dB (91%).

## 6. Circular Polarization

In order to achieve circular polarization requiring the radiation in the same direction of two perpendicular polarization components of equal amplitude but with a 90° phase difference, the object wave is changed from (21) to
(36)$$H_{obj}=H_{\phi_{0m}}\psi_{0m}(x,y){\;\hat{\phi }}_{0m}+jH_{\theta_{0e}}\psi_{0e}(x,y){\;\hat{\theta }}_{0e}$$
where (*θ*_0#_, *ϕ*_0#_) = (*θ*_0_, *ϕ*_0_) in (8) for this case, the latter being the angular coordinates of the common radiation direction shared by both polarization components. For the left-hand circular polarization (LHCP) of (36), the co- and cross-polar unit vectors are given by [29]
(37)$${\hat{\chi }}_{LHC}=\langle \begin{array}{c} c{\hat{o}}_{LHC}\\ x{\hat{p}}_{LHC} \end{array}\rangle =e^{\pm j\phi }\left(\;\hat{\theta }+\langle \pm \rangle j\;\hat{\phi }\right)/\sqrt{2}$$

By taking the dot products of the far-field vector function $G(\hat{r})$, presently that of the holographic LWA, expressed as
(38)$$G(\hat{r})=\hat{\theta }G_{\theta }(\hat{r})+\hat{\phi }G_{\phi }(\hat{r});\;\;G_{\langle \begin{array}{c} \theta \\ \phi \end{array}\rangle }=G\cdot \langle \begin{array}{c} \hat{\theta }\\ \hat{\phi } \end{array}\rangle$$
with the complex conjugates of the preceding unit vectors, the co- and cross-polar far-field functions are obtained as
(39)$$G_{{\langle \begin{array}{c} co\\ xp \end{array}\rangle }_{LHC}}(\hat{r})=G\cdot {\hat{\chi }}_{LHC}^{*}=\frac{e^{\langle \mp \rangle j\phi }}{\sqrt{2}}\left(G_{\theta }+\langle \mp \rangle jG_{\phi }\right)$$

For desired circular polarization, the axial ratio ℜ for a certain direction of far-field observation conveyed by $\;\hat{r}(\theta ,\phi )$ and the amplitudes of the co- and cross-polar fields towards that same direction are related by
(40)$$ℜ(\hat{r})=\left(\left|G_{co}\right|+\left|G_{xp}\right|\right)/\left(\left|G_{co}\right|-\left|G_{xp}\right|\right)$$
in which the far-field terms on the right side are from (39) for LHCP, although this relation is valid also for right-hand circular polarization (RHCP).

The cross-polar decoupling (or isolation), X*_isol_*, for any far-field direction $\hat{r}(\theta ,\phi )$ is defined as the ratio of the co-to cross-polar field amplitudes towards that direction, and is the reciprocal of the corresponding relative cross-polar level X*_rel_* with respect to the co-polar level, i.e.,
(41)$${Χ}_{isol}(\hat{r})=1/{Χ}_{rel}(\hat{r})=\left|G_{co}\right|/\left|G_{xp}\right|\overset{\mathrm{by}}{\underset{\left(40\right)}{=}}\left(ℜ+1\right)/\left(ℜ-1\right)$$
whereby the rightmost equation is found upon rearranging (40).

With the object wave of (36), the rest of the analysis remains the same as before for polarizations that are not necessarily circular. Synthesized according to the preceding object wave, a holographic LWA as depicted by Figure 14 for radiating a circular polarized main beam was designed for (*θ*_0_, *ϕ*_0_) = (20°, 45°). The simulated 3D absolute gain pattern is offered by Figure 15, while the planar pattern and the axial ratio ℜ, both in the *ϕ*_0_ plane, are presented in Figure 16 and Figure 17, respectively. Evident from Figure 16, strong main beams are produced towards the designated *θ*_0_ = 20° direction by the absolute gain pattern of Figure 16a as well as both theta and phi components of the gain as conveyed by Figure 16b, as required. An axial ratio of 1.09 dB (linear scale value of 1.134, close to unity) towards that direction is observed in Figure 17a simulated by CST and translated to an X*_isol_* of 24.042 dB via (41) as shown in Figure 17b. The corresponding co- and cross-polar patterns calculated by (39) using the simulated theta and phi gain components are given in Figure 18. As seen, the cross-polar radiation intensity towards the main beam is below that of the co-polar component by the considerable amount of 24 dB of *X_isol_*, as expected.

## 7. Sidelobe Suppression by Beam Cancellation

The herein-presented method for synthesizing multi-beam LWAs with any assortment of direction, amplitude, phase, and polarization of the beams arms us with all the tools necessary to achieve any desired radiation characteristics. The use of it to obtain circular polarization, a special case of dual-beam LWA, has just been presented. Another application is now showcased, and that is to suppress a targeted sidelobe (typically the most severe one) of, for example, a certain single-beam LWA. This may be achieved by incorporating onto the same hologram of the single-beam LWA two others that radiate beams towards the targeted sidelobe direction, which cancel out both in magnitude and phase the two orthogonal vector field components of the undesired lobe. This may thus be perceived as a multi-beam holographic LWA but with a specialized purpose—that of suppressing sidelobe radiation.

The showcased example is that of a TM-polarized single beam with main beam directed at *θ*_0*m*,*i*=1_ = 30°, *ϕ*_0*m*,*i*=1_ = 45°. The surface impedance distribution of the so-called originating first-stage hologram synthesized for radiating just this main beam is described by the first of the three sum terms within the inner parenthesis of the following.
(42)$$Z_{surf}^{3beam}=jX\left\{1+M\left[\begin{array}{l} H_{\phi_{0m,i=1}}\cos\left(\phi_{s}-\phi_{0m,i=1}\right)\times \cos\left\{k_{0}\left[\begin{array}{l} n\rho_{s}-\sin\theta_{0m,i=1}\\ \times \left(x\cos\phi_{0m,i=1}+y\sin\phi_{0m,i=1}\right) \end{array}\right]\right\}\\ +\left|H_{\phi_{0m,i=2}}\right|\cos\left(\phi_{s}-\phi_{0m,i=2}\right)\times \cos\left[\begin{array}{l} k_{0}\left[\begin{array}{l} n\rho_{s}-\sin\theta_{0m,i=2}\\ \times \left(x\cos\phi_{0m,i=2}+y\sin\phi_{0m,i=2}\right) \end{array}\right]\\ +∠H_{\phi_{0m,i=2}} \end{array}\right]\\ +\left|H_{\phi_{0m,i=2}}\right|\cos\left(\phi_{s}-\phi_{0m,i=1}\right)\times \cos\left[\begin{array}{l} k_{0}\left[\begin{array}{l} n\rho_{s}-\sin\theta_{0m,i=1}\\ \times \left(x\cos\phi_{0m,i=1}+y\sin\phi_{0m,i=1}\right) \end{array}\right]\\ +∠H_{\phi_{0m,i=3}} \end{array}\right] \end{array}\right]\right\}$$

In dB and linear scales, the black dotted traces in Figure 19(ai,aii), respectively, convey the co-polar patterns in the *ϕ*_0*m*,*i*=1_ = 45° plane of this originating hologram, in each of which a strong radiation towards the prescribed *θ*_0*m*,*i*=1_ = 30° is seen with a gain of 21.15 dBi (linear scale value of 130.2). Likewise, in those respective scales, the black dotted traces in Figure 19(bi,bii) again represent the co-polar patterns of this first-stage hologram, but this time in the *ϕ*_0*m*,*i*=2_ = 57° plane in which a TM-polarized sidelobe is incurred towards *θ*_0*m*,*i*=2_ = 38°, with a level of 13.1 dBi (linear scale value of 20.5). Plots in linear scales are included as they can portray differences in values more distinctly, as appreciated later when comparisons among the iterations are made.

The first refinement based on just the first two sum terms within the inner parenthesis of (42) is then carried out to eliminate this sidelobe. After determining $\left|H_{\phi_{0m,i=2}}\right|$ and $∠H_{\phi_{0m,i=2}}$ to be about 0.4 and 250°, respectively, that sidelobe is reduced to 7.3 dBi (5.3) upon synthesizing the two-beam hologram of this iteration, as the red traces of Figure 19(bi,bii) show. However, at the same time, the main beam is reduced to 20.9 dBi (122.9), as also conveyed by the red curves in Figure 19(ai,aii). Hence, a compensation of the main beam is needed to make up for this loss.

Constituting the next round of refinement, this is done by including this time, the third sum term of (42), of which the same amplitude $\left|H_{\phi_{0m,i=2}}\right|$ is utilized by this compensation term, now with a phase $∠H_{\phi_{0m,i=3}}$ of 50°. With this iteration comprising the three superposed holograms of (42), the main beam is observed by the blue traces of Figure 19(ai,aii) to indeed be compensated, boosting up to 21.69 dBi (147.7) and even exceeding the original 21.15 dBi. However, the sidelobe is also affected, as the blue traces of Figure 19(bi,bii) portray, although rising by only slightly to 7.9 dBi (6.16) from 7.3 dBi of the previous iteration. Importantly, this is still considerably less than the original level of 13.1 dBi. This study has thus demonstrated that sidelobes can be successful suppressed by the method of beam cancellation through the configurability of the beam direction, amplitude, phase, and polarization afforded by the present holographic multi-beam LWA.

In addition to the one provided above to reduce sidelobe levels, another typical method of sidelobe reduction is the use of a Hamming window [30]. Adopting such an approach, the results for sidelobe cancellations of a dual-beam holographic metasurface LWA, one TM-polarized beam and the other TE, are presented below. Simulated radiation patterns are presented for the cases prior to and after subjection to the Hamming window, according to the following expression:(43)$$0.54-0.46\times \cos(\frac{2\pi n}{N-1})\;\;\;0\leq n\leq N-1,$$

Starting with discussing the TM case, the co- and cross-polar radiation patterns (theta and phi gain components) are presented in Figure 20a,b below, in each of which the solid red line is for the original pattern before the Hamming window and the black dotted trace conveys the pattern upon the implementation of the Hamming window function stated above. It can be seen from Figure 20a that even though the co-polar main beams decrease by about 1.5 dB, the nearest-in co-polar sidelobe levels towards *θ* greater than the main beam’s 20° are observed to have slightly fallen. Howevcer, it is not these co-polar sidelobes, which are all at most about only 5 dBi, but rather the cross-polar ones that were originally serious at levels exceeding 10 dBi and thus in more crucial need of being alleviated. This is seen from the cross-polar pattern of Figure 20b, in which the most severe cross-polar sidelobe level of about 10.57 dBi towards (*ϕ*_TM,SL_ = 45°, *θ*_TM,SL_ = 25°) prior to Hamming window is seen to be mitigated to about 8.47 dBi upon the Hamming window, which shows a significant reduction of about 2.1 dB. Substantial mitigation towards most other theta directions about that 25° is also observable, even by as much as 6.8 dB towards *θ* = 17°. Although the sidelobe levels towards *θ* = 45° deteriorate, significant improvement towards most other theta directions of approximately 50°–90° is also distinct.

Coming to the TE polarization, the co- and cross-polar radiation patterns (phi and theta gain components, respectively) are presented in Figure 21a,b, with co-polar main beam directions toward (*ϕ*_TE_ = 135°, *θ*_TE_ = 35°), as conveyed by the former plot. The same explanations of the solid red and dotted black lines previously for the TM case reapply here for the TE case, being traces prior to and upon the Hamming window respectively. Before and after Hamming window, the co-polar main beam levels are 18.12 dBi and 16 dBi, respectively. In the Hamming window, the nearest-in sidelobe levels of the co-polar pattern are observed to have fallen by about 28 dB. For the cross-polar pattern of Figure 21b, substantial mitigation towards most theta directions is easily observable, and the cross-polar pattern with the most severe sidelobe towards (*ϕ*_TM,SL_ = 135°, *θ*_TM,SL_ = 30°) was reduced by about 2.64 dB. Additionally, the second most severe sidelobe towards (*ϕ*_TM,SL_ = 135°, *θ*_TM,SL_ = 25°) reduced by about 5.2 dB. It can thus be seen that using the Hamming window can mitigate most sidelobe levels so that the minor reduction of the co-polar main beam can be neglected.

## 8. Measurements on Manufactured Prototype

Herein presented are reports of the measurement results obtained from experiments carried out on a manufactured holographic LWA that radiates multiple beams of various mutually orthogonal principal polarizations with tailorable relative gains designed according to our approach. Particularly, the second of the two cases investigated by simulations earlier was fabricated, with $\left(\theta_{0m}=35^{\circ },\phi_{0m}=45^{\circ }\right)$, $\left(\theta_{0e}=20^{\circ },\phi_{0e}=135^{\circ }\right)$, and (*A^TM^* = 0.667, *A^TE^* = 0.333).

A via hole is drilled through the phase reference point (coordinate origin) of the prototype hologram, through which the inner conducting pin of a 2.92 mm panel socket jack coaxial probe connector passes and protrudes out perpendicularly from the plate, as schematized by Figure 22, serving as a monopole source which radiates the reference cylindrical waves.

As per the simulations in the previous section, the manufactured antenna uses a Rogers 5880 PCB board, the top side and SMA soldering of which are photographed in Figure 23a,b, respectively. The 2.92 mm panel jack connector, with an operation frequency from 18 GHz to 40 GHz that encompasses our 28 GHz, was welded onto the underside ground plane, as the photo of Figure 23b shows, and its inner copper pin tunnels through the board to protrude vertically out on the other (upper) side.

The process of the sample fabrication is described next. Firstly, we provided a Gerber file of our design in CST to a company, Cheer-Time Enterprise Co., Ltd., located at No. 311, Qionglin S. Rd., Xinzhuang Dist., New Taipei City 242067, Taiwan (R.O.C.), which is then checked by an engineer there. Upon clearance, the board will enter the production process. The first step involves the cutting of the dielectric sheets, specifically the Rogers RT5880, that were provided to the company. According to the work order, the company cuts the substrate into the required size of the working panel. The next step is dry film lamination, whereby the board is pre-processed to clean and slightly etch the surface of the board, which is then sent to a laminator for lamination. This is followed by exposure and development, whereby the surface of the board is covered with a layer of film, and upon exposure, the required pattern is transferred to the board surface. After development, pattern plating ensues, whereby secondary copper and tin-lead plating increase the thickness of the outer circuit. Tin plating is to protect the copper surface under the tin surface from being dissolved by the etching liquid. After etching, NC-routing follows according to the requirements of the required outline drawing. The shape is processed and cut to produce the size that meets our design needs. Finally, electrical testing is carried out, followed by final visual inspection.

We used the antenna measurement platform (model SB10110-45) developed by the company, Taiwan Microwave Circuit Co., Ltd., located at 3F., No. 72, Chenggong 10th St., Zhubei City, Hsinchu County 302050, Taiwan (R.O.C.), to measure the antenna radiation field pattern. The technology used by the company’s antenna measurement machine is near-field measurement technology combined with the algorithm developed by the company to obtain far-field data for generating antenna patterns. Radiation patterns of a WR-28 standard [31] gain horn measured by this equipment have also been compared with those from external antenna measurements using far-field technology, as shown in Figure 24. Good agreement with traditional far-field measurements is observed.

Figure 25 shows the photograph of the fabricated holographic LWA braced in its experimental setup position and surrounded by absorbers inside an anechoic chamber for measurements of far-field radiation patterns at the designated 28 GHz.

The simulated and measured reflection coefficients S_11_ over a band about the operation frequency is conveyed by Figure 26. Good impedance matching throughout the band centered at 28 GHz achieved by the simulated design is verified experimentally.

Shown in Figure 27 are the simulated and measured co- and cross-polar gain patterns of the manufactured holographic antenna that radiates two beams with mutually orthogonal polarization. It can be seen from Figure 27a that the simulated and measured co-polar gain (theta component) patterns of the TM polarized beam agree well with each other, with a maximum level of radiation achieved accurately towards the prescribed main beam direction of (*θ*_0m_ = 35°, *ϕ*_0m_ = 45°) of nearly 20 dBi for both. The corresponding patterns (simulated and measured patterns) of the cross-polar (phi) component towards that main beam direction are seen from Figure 27b to be indeed much lower than the co-polar intensity by around 10 dB. For the other beam, which is TE polarized, the consistency between the measured and simulated co-polar gain patterns (phi component) is evident from Figure 27c, while the agreements of the cross-polar (theta) pattern are observed in Figure 27d. The co-polar main beam level of around 12 dBi is accurately reproduced in the experiment towards the designated direction of *θ*_0e_ = 20° and *ϕ*_0e_ = 135° and is considerably stronger than the cross-polar level of about 0 dBi. In addition, the difference of about 6 dB between the intensities of the two measured co-polar main beams is also consistent with expectation by theory.

## 9. Comparisons with Existing Literature

Finally, the capabilities of the proposed holographic LWA are compared to those of previous works in Table 1. In particular, a total of 18 other papers are held up in terms of eight aspects, such as beam multiplicity and its directional dimensionality, gain amplitude, phase, and polarization controls, among a few others. It is observed that just short of half of them (8, to be precise) are able to achieve multiple beams; 14 of the 18 are able to emit beams on any azimuthal plane, but 9 of these 14 are just single-beam antennas. Moreover, only seven works have reported amplitude control of the radiated beam. Only [16,19,20,21,27,32] possess capabilities in beam multiplicity and azimuthal variability. Among these, only [16,21,27] are unable to provide any forms of beam control, with the rest ([19,20,32]) all afford gain, polarization, and phase controls. Among all 18 contenders, only seven of the 18 claimed the capability of beam polarization control, most of them having used the method of tensor impedance to synthesize holograms that achieve the desired effects, and all of them offered phase control as well. However, only [32] has used hologram partitioning to achieve the control of polarization and phase, calling for multiple sources. Of these, 12 of the 18 do not require partitioning of the hologram. None of these contemporaries features sidelobe suppression and compensation, except for [26], which offers a method of sidelobe cancellation. Finally, almost half of them (nine, to be exact) have reported experimental verifications by measurements of fabricated prototypes.

Our present work is able to achieve all eight aspects. Furthermore, the greatest advantage of ours is that due to the simplicity afforded by the commonly realizable reference wave, only scalar impedance elements—as opposed to overly complicated tensor ones—are needed, a fact that we have proven upon formulation from the first principles of tensor impedance concepts. This allows for far simpler procedures for designing holographic LWAs.

Some additional quantitative data for several papers are tabulated in Table 2 to facilitate the comparisons, such as the realized gain and side-lobe levels (SLL).

Further quantitative measurement data in comparison with all papers in Table 1 are also offered in Table 3.

## 10. Comparison of Holographic Beamforming Arrays Afforded by the Present Design with Conventional Phased Arrays and Sector Antennas

As we approach the era of 6G communications, although enhanced data speeds and capacities can be anticipated, systems are also more susceptible to losses arising from the propagation and penetration of the signal waves through the air and obstacles. This can be alleviated by utilizing large beamforming antenna arrays for higher directivities and narrower beamwidths. Due to the mobility of the users, the beamformer array is required to scan and steer its beams to serve numerous users at speeds high enough to be undiscernible to the users. This then points at electronically scanned phased arrays (ESPA) as an essential mainstay for past and present generations of wireless communications. ESPAs are, however, high in cost, size, weight, and power consumption. On the contrary, holographic beamforming arrays (HBFA), realizable by our herein proposed holographic LWA design, are considerably lower in all those aspects.

In terms of beamforming architecture, beams of both ESPA and HBFA are generated and steered by the radiation or reception of the RF signal from or by many elements. There are, however, two main differences, namely the way in which the signals are steered and the placements of the amplification. At every element in ESPAs, two types of devices are located: phase shifters for prescribing the required phase and either variable gain amplifiers or attenuators to control the gain, correct amplitude errors, or offset RF losses. However, in HBFAs, antenna elements within a row are coupled to a feed line and a tuning element is attached to each of them for adjusting the relevant electrical parameter, thereby modulating the signal propagation along the feed line. In turn, this allows the coupling to each antenna element to be configured in such a way that its amplitude and phase are controlled to achieve beam steering and forming. As this process is analogous to the synthesis of a hologram, it is thus called holographic beam forming, and because no active amplification is needed, HBFAs are deemed as passive. Although an HBFA has a similar, if not larger, number of elements compared to an ESPA to achieve the same directivity, the former entails only a single low-cost, discrete RF component at each element, whereas the ESPA requires many such components to be associated with each antenna element. Hence, an HBFA is considerably cheaper and less energy-consuming than a comparable ESPA. With wider scan ranges, the number of base stations can be reduced.

Where basic performances are concerned, ESPAs and HBFAs are comparable. They are able to steer and scan beams as rapidly as they switch between transmit and receive modes of operation. Both are capable of fulfilling typical EIRP (effective isotropic radiated power) and G/T (receiver gain to system noise temperature) criteria of transmission and reception operations. Achievable bandwidths of the two technologies are also similar. However, important differences still lie between these two types of arrays. Particularly, the ±80° range of elevation scan angles in any given azimuth plane typically attainable by HBFAs is generally wider than the ±60° span of ESPAs. This is attributed to the broader beam patterns associated with the smaller subwavelength-sized elements of HBFAs as compared to the larger typically half-wavelength sized ones of ESPAs.

The cost of beamforming already constitutes a significant portion of the total cost of a communications system. With the increasing number of elements and the associated hardware required of future 6G networks, beamforming costs will be a critical factor for mobile operators. Evidently, hardware costs take up a large part of the financial burden. In general, aside from being exempted from the need for phase shifters, HBFAs call for just one amplifier for transmission and one for reception. On the other hand, ESPAs require a large number of both kinds of devices. Furthermore, the larger number of devices in ESPAs also translates to higher power consumption and thus higher operation costs compared to HBFAs. The need for more space around the many power-hungry devices of ESPAs for heat dissipation also renders them heavier and bulkier than their HBFA counterparts.

Commonly mounted on towers, sector antennas are used in traditional and contemporary communications systems. Rather than the structural form of them, it is the sectoral shape of the radiation pattern of this type of antenna which its name expresses. Physically, they tend to be elongated or cylindrical instead. Multiple sector antennas with different angular spans of sectoral regions, typically 60°, 90°, or 120°, are usually hosted by a single tower. The overlap between adjacent sector beams of two different sector antennas cannot be too small or too large; the former results in improper handover, whilst the latter leads to excessive interference among users. Moreover, the wide sectoral span of each beam is intended to serve all users within that sector regardless of their number or location in a somewhat “sweeping” but “mindless” way that is energy wasteful, low in data capacity, weak in signal transmission, and narrow in bandwidth. On the contrary, HBFAs offer cognitively configured narrow and precisely directed beams that dedicatedly serve specific users with much less waste, higher throughputs, stronger directivities, and better spectral efficiencies. The risks of interference and handover drop-offs are also lower in HBFAs as compared to sector antennas.

## 11. Conclusions

Through derivation from fundamentals of the tensor surface impedance theorem, this work has proven for practical reference beam sources the valid use of only simple scalar impedance elements for the design and synthesis of holographic leaky-wave antennas composed of flat metasurfaces that are capable of radiating simultaneous multiple beams with tailorable magnitudes and all at a single frequency towards any prescribed directions in both elevation and azimuth (not restricted to be within a common plane) without the need to partition the hologram or the use of multiple sources. The practical design implementation is demonstrated at a central frequency of 28 GHz. Aside from the robustness of generating non-coplanar beams, the individual control of the amplitude, phase, and polarization of each beam, all under any one operation scenario and without hologram partitioning, are all achievable by using just scalar impedance elements, thereby dramatically simplifying a design procedure which otherwise would be excessively complicated, cumbersome, and slow had tensor elements been entailed instead. This constitutes a major strength of our work. Not only were the designs of dual and quadruple beams with different polarizations showcased, the feasibility of sidelobe elimination was also successfully demonstrated. The prototype of the dual-beam case was manufactured and measured. Experimental results agree well with simulations in terms of the impedance matching bandwidth and far-field performance.

At present, our holographic antenna demonstrates one specific radiation pattern at a time and if another pattern is sought, a different interferogram has to be synthesized. Injecting reconfigurability for greater applicability is important. This points at a research topic called reconfigurable intelligent surfaces (RIS), which has recently gained importance and attracted considerable attention. These surfaces are textures made of metasurfaces comprising subwavelength elements called meta-atoms that use electronic circuit devices to reconfigure the surface properties so as to alter the mobile wireless propagation environment or tailor the radio channel for enhanced capacities, throughputs, data speeds, bandwidths, energy and spectral efficiencies, accuracies, and reduced interferences. In the context of our proposed holographic LWA, the original reference surface wave excitation can be replaced by the impingent plane wave, while the intended object waves to be reconstructed are left as they were before. In this way, the spirit of RIS is fulfilled as the beam patterns reradiated by the RIS can be tailored in ways such that traditional Snell’s laws are bent and the wireless channels are optimized. For achieving reconfigurability, literary works have reported the use of diodes, varactors, PINs, liquid crystals, MEMS, CMOS among others to alter surface properties such as the reflection phase, effective refractive index, and surface impedance. Hence, the future prospects of the holographic LWA proposed herein are bountiful.

## Figures and Tables

**Figure 1 sensors-24-02422-f001:**
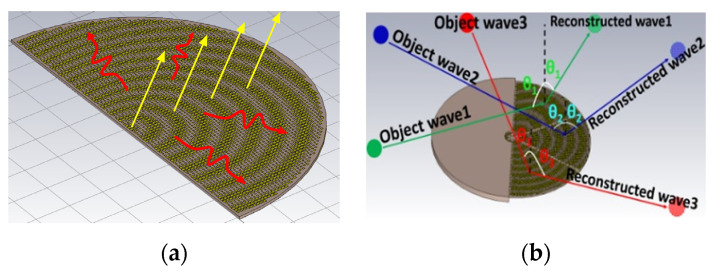
(**a**) Surface waves (undulating arrows) excited on an artificial impedance surface that scatter waves by changes in the surface impedance to produce the desired radiation beams (straight arrows); (**b**) image reconstruction of objects comprising distributed sources.

**Figure 2 sensors-24-02422-f002:**
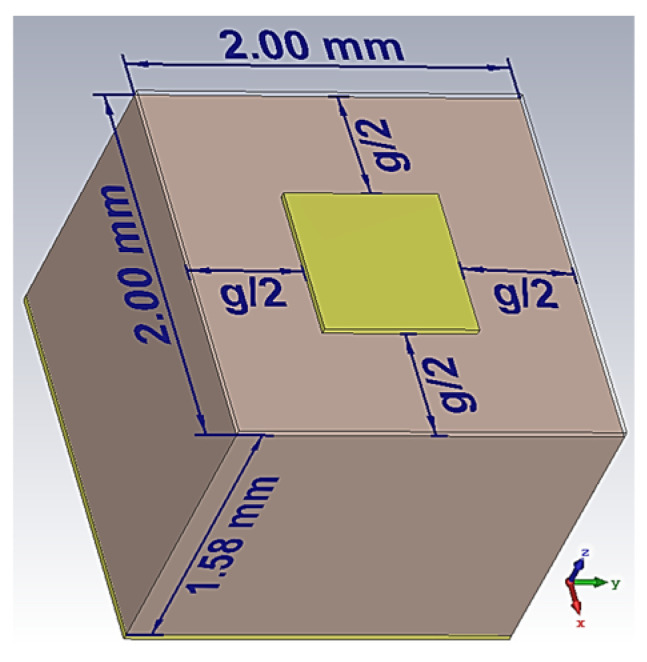
Schematic of the 2×2×1.575
${\mathrm{mm}}^{3}$ unit cell with a variable gap size *g*.

**Figure 3 sensors-24-02422-f003:**
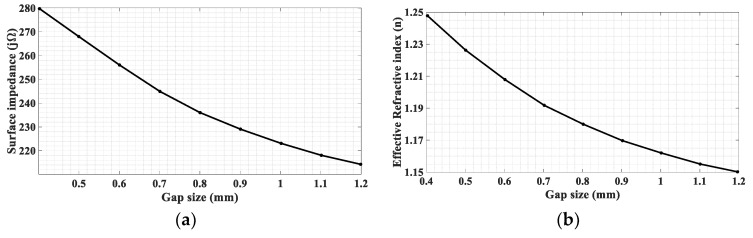
(**a**) Surface impedance *Z_patch_* of the unit cell as a function of gap size; the range of gap size is from 0.4 mm to 1.2 mm. (**b**) Effective refractive index *n* of the unit cell as a function of gap size; the range of gap size is from 0.4 mm to 1.2 mm.

**Figure 4 sensors-24-02422-f004:**
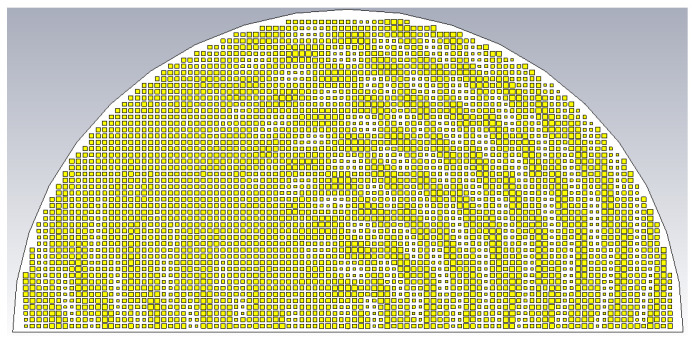
Holographic pattern base on interferences that generate dual beams at θ_0*m*_ = 20°, *ϕ*_0*m*_ = 45° (TM) and *θ*_0*e*_ = 50°, *ϕ*_0*e*_ = 135° (TE).

**Figure 5 sensors-24-02422-f005:**
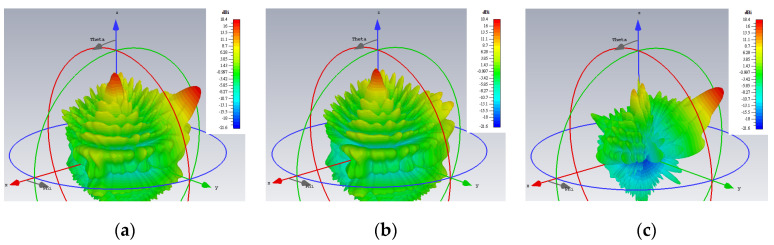
Simulated 3D far field gain patterns: (**a**) absolute gain; (**b**) theta component of the *E*-field gain with beam towards *θ*_0*m*_ = 20°, *ϕ*_0*m*_ = 45° (TM); and (**c**) phi component of the *E*-field gain with beam towards *θ*_0*e*_ = 50°, *ϕ*_0*e*_ = 135° (TE).

**Figure 6 sensors-24-02422-f006:**
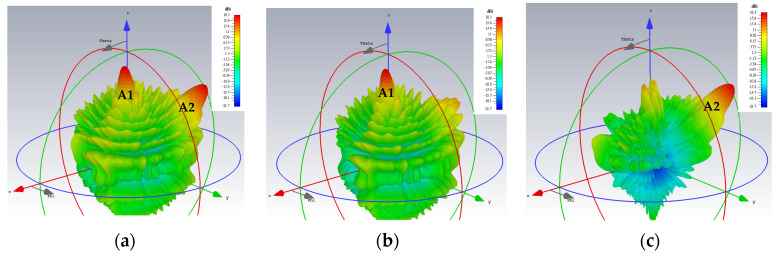
Simulated 3D far field gain patterns for dual-beam, dual-polarized antenna with *A^TM^* = *A^TE^* = 0.5: (**a**) absolute gain; (**b**) theta component of *E*-field gain with beam towards *θ*_0*m*_ = 15°, *ϕ*_0*m*_ = 45° (co-polar for TM); and (**c**) phi component of *E*-field gain with beam towards *θ*_0*e*_ = 45°, *ϕ*_0*e*_ = 135° (co-polar for TE).

**Figure 7 sensors-24-02422-f007:**
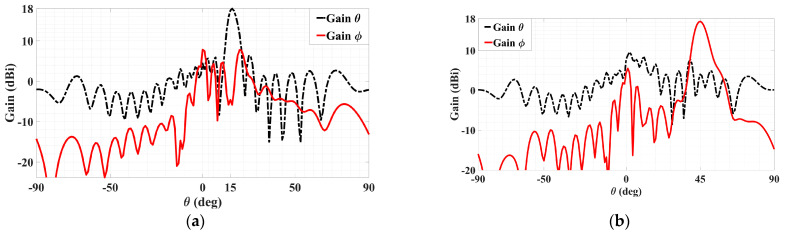
Simulated planar far field patterns for dual-beam, dual-polarized antenna with *A^TM^* = *A^TE^* = 0.5 in (**a**) *ϕ* = *ϕ*_0*m*_ = 45° plane, showing strong radiation of co-polar theta component but weak cross-polar phi towards *θ*_0*m*_ = 15° and (**b**) *ϕ* = *ϕ*_0*e*_ = 135° plane, showing strong radiation of co-polar phi component but weak cross-polar theta towards *θ*_0*e*_ = 45°.

**Figure 8 sensors-24-02422-f008:**

Simulated 3D far field gain patterns for dual-beam, dual-polarized antenna with *A^TM^* = 0.333, *A^TE^* = 0.667: (**a**) absolute gain; (**b**) theta component of *E*-field gain with beam towards *θ*_0*m*_ = 15°, *ϕ*_0*m*_ = 45° (co-polar for TM); and (**c**) phi component of *E*-field gain with beam towards *θ*_0*e*_ = 45°, *ϕ*_0*e*_ = 135° (co-polar for TE).

**Figure 9 sensors-24-02422-f009:**
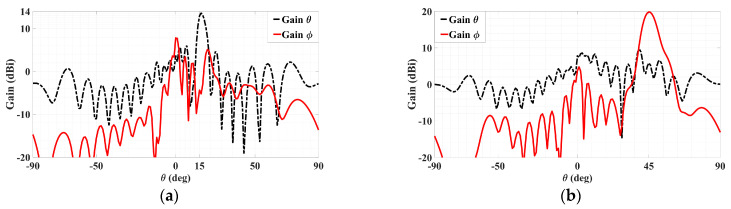
Simulated planar far field patterns for dual-beam, dual-polarized antenna with *A^TM^* = 0.333, *A^TE^* = 0.667, in (**a**) *ϕ* = *ϕ*_0*m*_ = 45° plane, showing strong radiation of co-polar theta component but weak cross-polar phi towards *θ*_0*m*_ = 15°, and (**b**) *ϕ* = *ϕ*_0*e*_ = 135° plane, showing strong radiation of co-polar phi component but weak cross-polar theta towards *θ*_0*e*_ = 45°. Maximum 20 dBi strength of phi component in is (**b**) about 20log_10_(2) = 6 dB stronger than the 14 dBi strength of theta component in (**a**).

**Figure 10 sensors-24-02422-f010:**
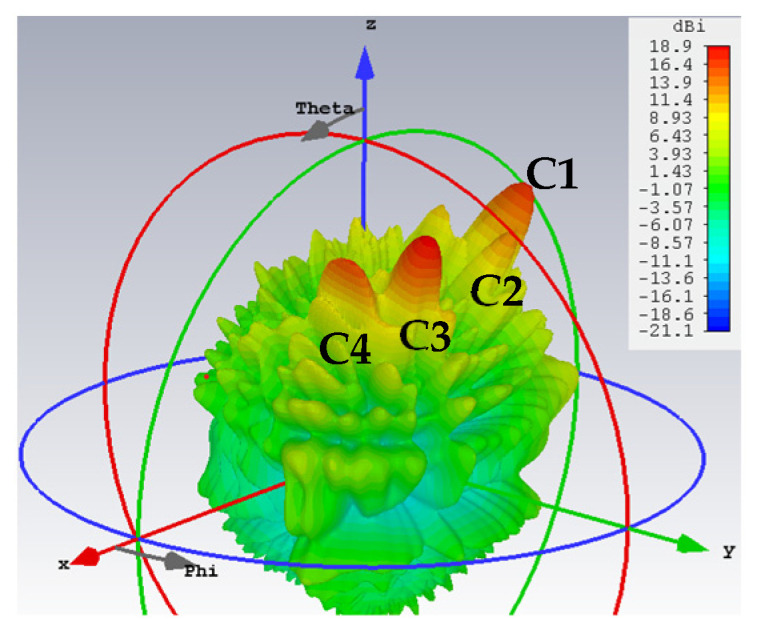
Simulated 3D radiation pattern of all-TM-polarized quadruple-beam holographic antenna for (*ϕ*_0*m*,1_ = 150°, *ϕ*_0*m*,2_ = 120°, *ϕ*_0*m*,3_ = 60°, *ϕ*_0*m*,4_ = 30°) and (*θ*_0*m*,1_ = *θ*_0*m*,2_ = *θ*_0*m*,3_ = *θ*_0*m*,4_ = 30°) with (*A*_1_^TM^ = *C*_1_ = 0.2, *A*_2_^TM^ = *C*_2_ = 0.15, *A*_3_^TM^ = *C*_3_ = 0.4, *A*_4_^TM^ = *C*_4_ = 0.25).

**Figure 11 sensors-24-02422-f011:**
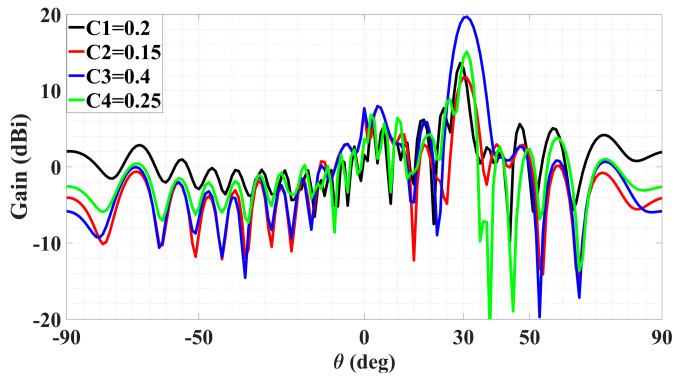
Simulated gain pattern of all-TM-polarized quadruple-beam holographic LWA of Figure 10. All beams converge towards a common *θ*_0*m*_ = 30°.

**Figure 12 sensors-24-02422-f012:**
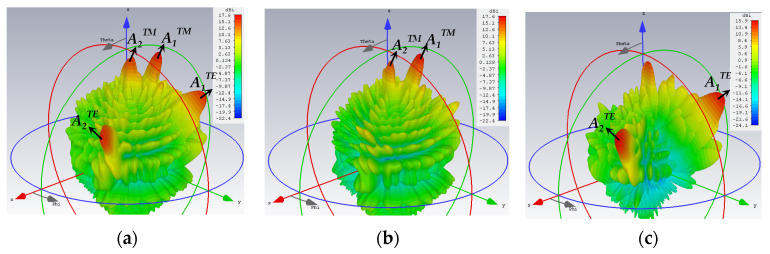
Simulated 3D far field gain patterns for quad-beam LWA, with two TM- and two TE-polarized beams with (*A*_1_^TM^ = *A*_2_^TM^ = *A*_1_^TE^ = *A*_2_^TE^ = 0.25): (**a**) absolute gain; (**b**) theta component of gain with two strong beams towards (*θ*_0*m*,1_ = 15°, *ϕ*_0*m*,1_ = 150°) and (*θ*_0*m*,2_ = 5°, *ϕ*_0*m*,2_ = 60°) (co-polar for TM); and (**c**) phi component of gain with beams towards (*θ*_0*e*,1_ = 45°, *ϕ*_0*e*,1_ = 120°) and (*θ*_0*e*,2_ = 55°, *ϕ*_0*e*,2_ = 30°) (co-polar for TE).

**Figure 13 sensors-24-02422-f013:**
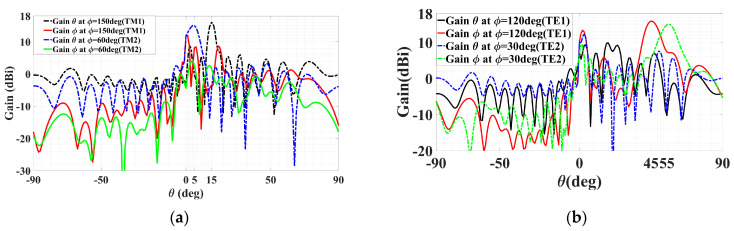
Simulated planar far field patterns for quad-beam LWA, with two TM- and two TE-polarized beams, (*A*_1_^TM^ = *A*_2_^TM^ = *A*_1_^TE^ = *A*_2_^TE^ = 0.25), in various phi planes containing the beams, (**a**) TM beams in 150° and 60° phi-cuts, showing strong radiation of co-polar theta component but weak cross-polar phi towards *θ*_0*m*,1_ = 15° and *θ*_0*m*,2_ = 5°, and (**b**) TE beams in 120° and 30° phi planes, showing strong radiation of co-polar phi component but weak cross-polar theta towards *θ*_0*e*,1_ = 45° and *θ*_0*e*,2_ = 55°.

**Figure 14 sensors-24-02422-f014:**
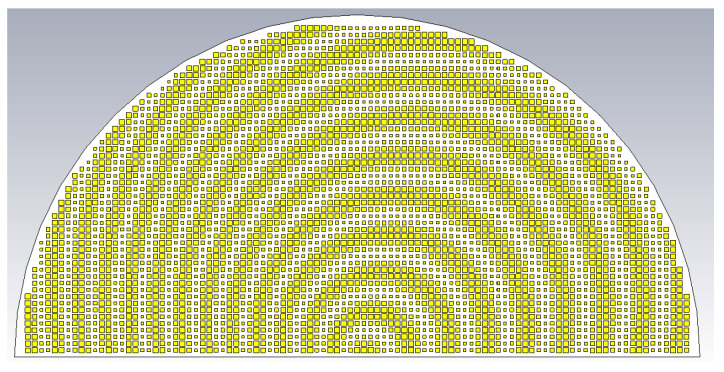
Hologram as LWA for radiating circular polarized beam towards (*θ*_0_, *ϕ*_0_) = (20°, 45°).

**Figure 15 sensors-24-02422-f015:**
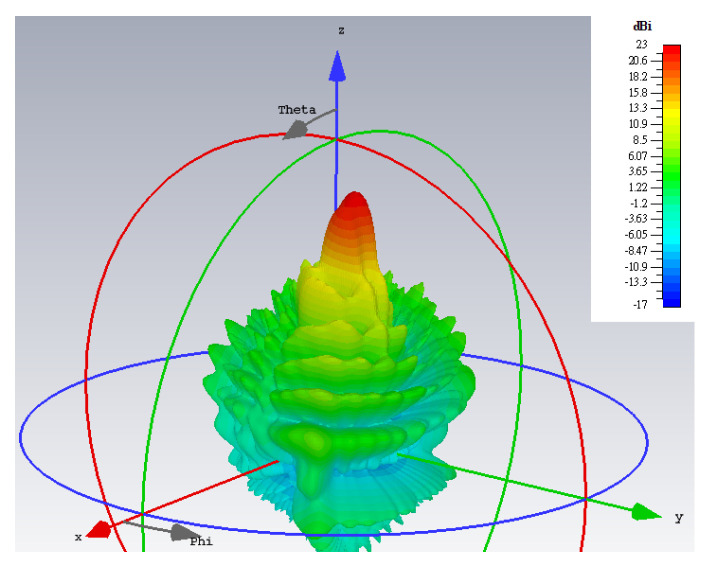
Simulated 3D far field absolute gain pattern with circular polarized main beam towards (*θ*_0_, *ϕ*_0_) = (20°, 45°).

**Figure 16 sensors-24-02422-f016:**
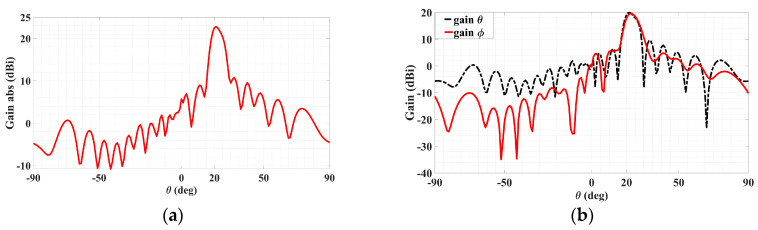
Radiation patterns in *ϕ*_0_ = 45° plane of circular-polarized LWA for (**a**) absolute gain and (**b**) in the same plot, theta and phi components of gain.

**Figure 17 sensors-24-02422-f017:**
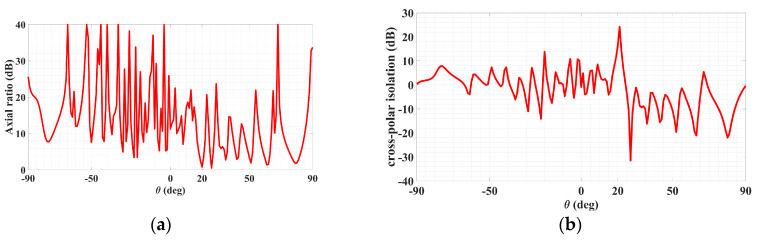
Radiation patterns in *ϕ*_0_ = 45° plane of circular-polarized LWA for (**a**) absolute gain and (**b**) in the same plot, theta and phi components of gain.

**Figure 18 sensors-24-02422-f018:**
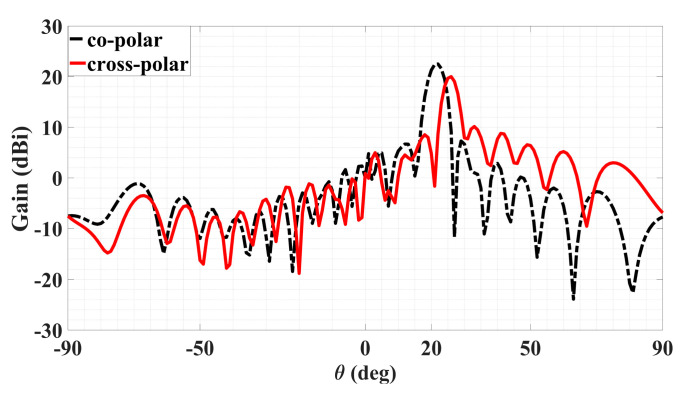
Co- and cross-polar far-field patterns of circular polarized LWA in *ϕ* = *ϕ_inc_* plane for main beam towards *θ*_0_ = 20° in CST.

**Figure 19 sensors-24-02422-f019:**
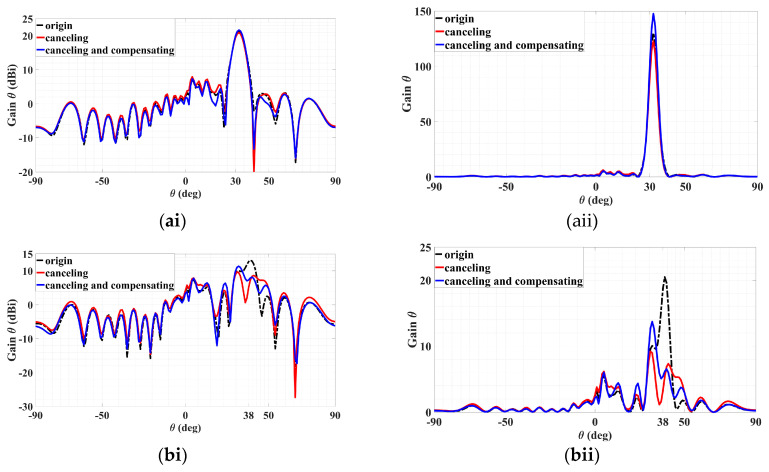
The above figure is a composite diagram of original, sidelobe cancellation, sidelobe cancellation, and main beam compensation and is divided into (**a**) main beam (TM) co-polarization; (**ai**) dB scale, (**aii**) linear scale, and (**b**) sidelobe TM co-polarization; (**bi**) dB scale, (**bii**) linear scale.

**Figure 20 sensors-24-02422-f020:**
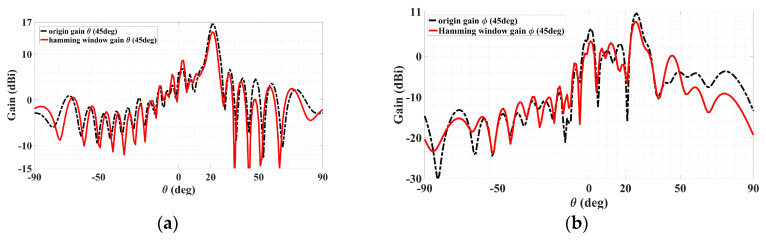
Radiation patterns in *ϕ*_TM_ = 45° plane containing TM polarized beam, with red and black dotted traces for cases before and after the Hamming window. (**a**) Co-polar pattern with main beam towards (*ϕ*_TM_ = 45°, *θ*_TM_ = 20°) and (**b**) cross-polar pattern with the most severe sidelobe towards (*ϕ*_TM,SL_ = 45°, *θ*_TM,SL_ = 25°), reduced by about 2.1 dB.

**Figure 21 sensors-24-02422-f021:**
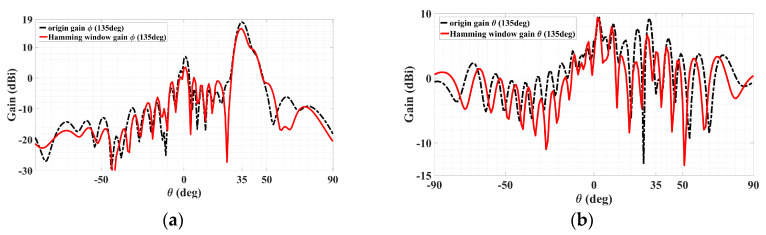
Radiation patterns in *ϕ*_TE_ = 135° plane containing TE polarized beam, with red and black dotted traces for cases before and after the Hamming window. (**a**) Co-polar pattern with main beam towards (*ϕ*_TE_ = 135°, *θ*_TE_ = 35°) and (**b**) cross-polar pattern with the most severe sidelobe towards (*ϕ*_TM,SL_ = 135°, *θ*_TM,SL_ = 30°), reduced by about 2.64 dB.

**Figure 22 sensors-24-02422-f022:**
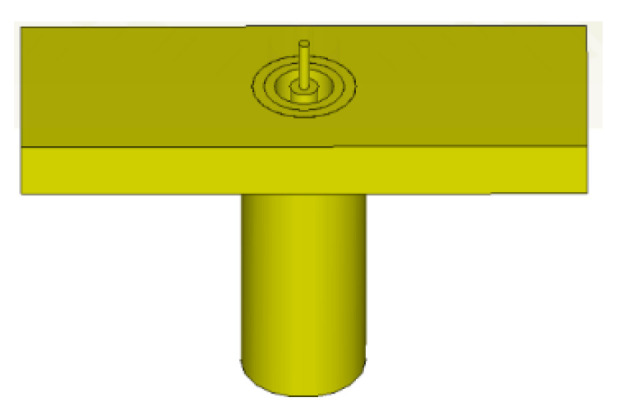
Schematic of a 2.92 mm panel jack coaxial probe connector.

**Figure 23 sensors-24-02422-f023:**
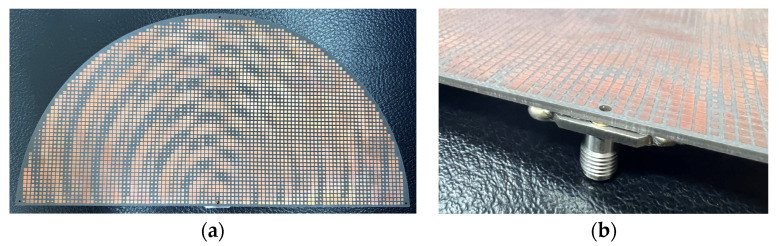
(**a**) Top and (**b**) SMA soldering views of the holographic antenna fabricated in θ = 45° and 135° modes.

**Figure 24 sensors-24-02422-f024:**
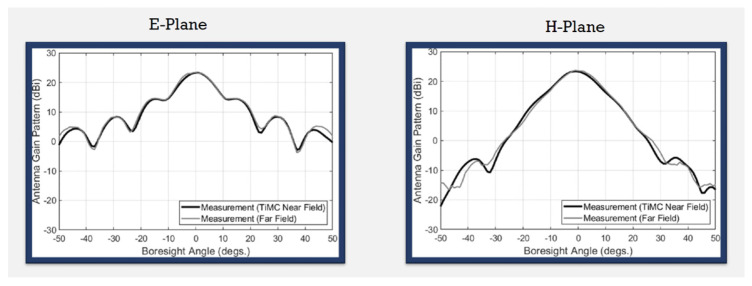
Comparison of measured radiation patterns of WR-28 standard horn antenna obtained by the near-field measurement technology of Taiwan Microwave Circuit Company with those obtained by measurements in far-field anechoic chamber.

**Figure 25 sensors-24-02422-f025:**
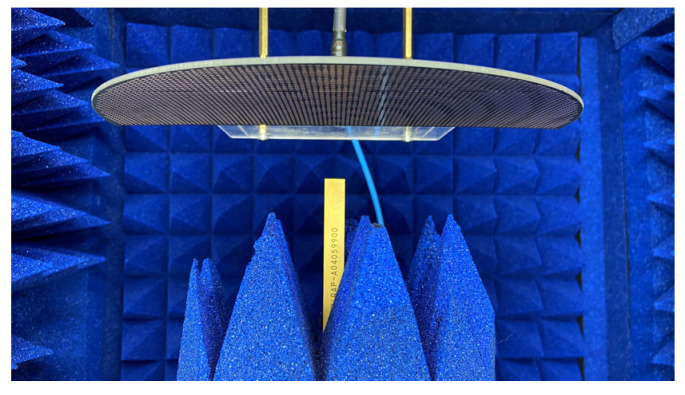
Measurement environment of 28 GHz holographic antenna in Taiwan Microwave Circuit company.

**Figure 26 sensors-24-02422-f026:**
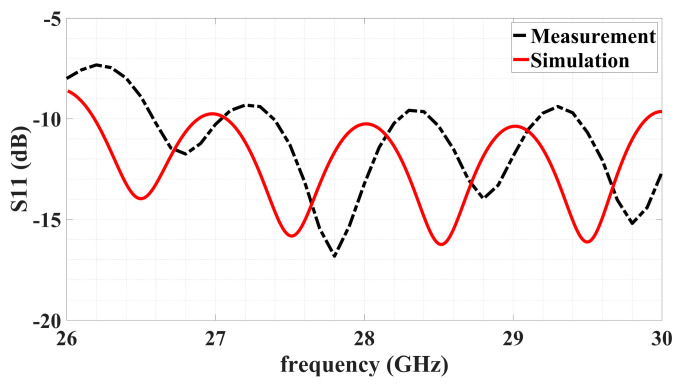
Variation with frequency of simulated and measured S_11_ over band centered around 28 GHz operating frequency.

**Figure 27 sensors-24-02422-f027:**
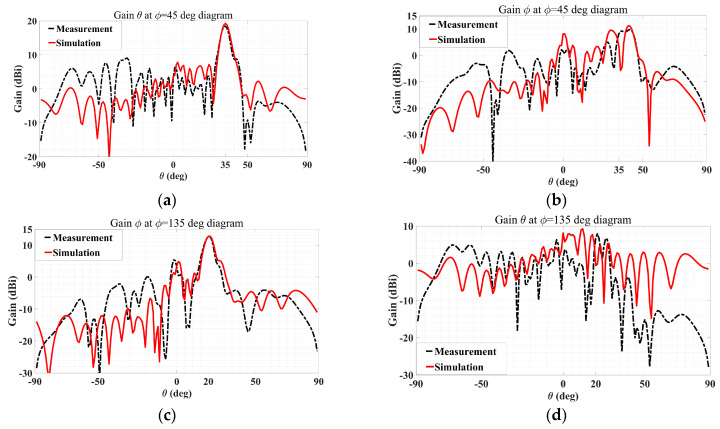
Measured and simulated gain patterns of dual-polarization multiple beams holographic LWA with (*A*^TM^ = 0.667, *A*^TE^ = 0.333): (**a**) co-polar and (**b**) cross-polar in *ϕ*_0m_ = 45° plane with *θ*_0m_ = 35°; (**c**) Co-polar and (**d**) Cross-polar in (**b**) *ϕ*_0e_ = 135° plane with *θ*_0e_ = 20°.

**Table 1 sensors-24-02422-t001:** Comparison of millimeter-wave holographic LWA.

Reference	Multiple Beams	Beams in Any Phi Plane	Gain Amplitude Control	Beam Polarization Control	Beam Phase Control	No Hologram Partitioning	Sidelobe Suppression and Compensation	Contains Measurement Results
[1]	✗	✓	✗	✓	✓	✓	✗	✗
[3]	✗	✓	✗	✗	✓	✓	✗	✗
[12]	✗	✓	✗	✗	✗	✓	✗	✗
[13]	✗	✓	✗	✗	✗	✓	✗	✓
[14]	✗	✓	✗	✓	✓	✗	✗	✓
[15]	✗	✓	✗	✗	✗	✗	✗	✓
[16]	✓	✓	✗	✗	✗	✗	✗	✓
[17]	✗	✓	✓	✓	✓	✓	✗	✗
[18]	✗	✓	✓	✓	✓	✓	✗	✓
[19]	✓	✓	✓	✓	✓	✓	✗	✗
[20]	✓	✓	✓	✓	✓	✓	✗	✗
[21]	✓	✓	✓	✗	✗	✓	✗	✓
[22]	✓	✗	✓	✗	✗	✓	✗	✓
[23]	✓	✗	✓	✗	✗	✓	✗	✗
[26]	✗	✓	✗	✗	✓	✗	✗	✗
[27]	✓	✓	✗	✗	✗	✗	✗	✗
[28]	✗	✗	✗	✗	✗	✓	✗	✓
[32]	✓	✓	✓	✓	✓	✗	✗	✓
Present work	✓	✓	✓	✓	✓	✓	✓	✓

**Table 2 sensors-24-02422-t002:** Comparison of quantitative data with papers in literature.

Ref.	Freq (GHz)	Area of the Aperture	Number of Ports	Number of Beams	Realized Gain (dBi)	SLLs (dB)
[19]	24	$\pi \times {\left(10\lambda_{0}\right)}^{2}$	2	2	21.5	9
[21]	14	$10\lambda_{0}\times 10\lambda_{0}$	7	7	20	10
[32]	12	$9.6\lambda_{0}\times 8\lambda_{0}$	2	2	16	6
This work	28	$0.5\pi \times {\left(9.4\lambda_{0}\right)}^{2}$	1	1TM + 1TE	18 @ for TM 17.3 for TE	7.84 for TM 9.61 for TE

**Table 3 sensors-24-02422-t003:** Comparison of measurement data with papers in the literature.

Reference	$S_{11}$ (Meas.)	Co-Pol (Meas.)	Cross-Pol (Meas.)
[1]	✗	✗	✗
[3]	✗	✗	✗
[12]	✗	✗	✗
[13]	−25 dB (17 GHz) −17 dB (20 GHz)	16.7 dB (17 GHz) 20.5 dB (20 GHz)	✗
[14]	✗	15 dB (12 GHz)	5 dB (12 GHz)
[15]	−18 dB (23 GHz) (sim.)	15 dB (23 GHz)	✗
[16]	✗	−5 dB (16 GHz) (normalized)	✗
[17]	✗	✗	✗
[18]	✗	28 dBi (8.4 GHz)	✗
[19]	✗	✗	✗
[20]	✗	✗	✗
[21]	−22.5 dB (14 GHz)	20 dB (14 GHz)	10 dB (14 GHz)
[22]	✗	12 dB (18 GHz)	✗
[23]	✗	✗	✗
[26]	✗	✗	✗
[27]	✗	✗	✗
[28]	−9 dB (10 GHz)	14.7 dB (10 GHz)	−22 dB (10 GHz)
[32]	✗	15 dB (12 GHz)	7.5 dB (12 GHz)
Present work	−13 dB (28 GHz)	19 dB (TM) 13 dB (TE)	10 dB (TM) 9 dB (TE)

## Data Availability

The data supporting this study are included within the article.

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
