# Peer review of "Tensor-Free Holographic Metasurface Leaky-Wave Multi-Beam Antennas with Tailorable Gain and Polarization"

_sensors, 2024, doi:10.3390/s24082422_

Round 1

Reviewer 1 Report

Comments and Suggestions for Authors

General substantive comments:

1. The article seems methodologically correct in the context of synthesis of the selected antenna design. Unfortunately, there is no relation to the topic of the special issue and the essence of Sensors journal. As it is, the article is more suitable for a magazine related to antenna technology. It should be noted, that the proposed antenna design can be supplemented with content of communication, sensing and localization in 6G systems. The background of the application should be presented. Then make the link to the holographic antennas and complete the text with the design assumptions (in the considered 28 GHz frequency band) that will be important in relation to the problem of communication, sensing and objects localization in 6G systems.

2. What are the advantages/disadvantages of the proposed design in relation to phased antenna arrays? The proposed solution should be referred to the design and operation of phased antenna arrays considering i.e. the problem of sensing/objects localization in 6G systems.

3. What are the advantages/disadvantages of the proposed design (e.g. multiple beams holographic LWA) in relation to sector antennas? This issue should be discussed similarly to Note 2.

4. Please describe the technological process of sample fabrication.

5. Please describe the equipment, test stands, and measurement methods used in the experimental studies.

Editorial corrections:

a) as a first step, the formatting of variables should be unified to a contemporary standard. This means, that all symbols used in text/equations/figures/tables should be formatted uniformly: italics for scalar variables, bold non-italic for matrices/vectors in the whole manuscript. In its current form, the mathematical notation is not very clear;

b) all symbols should be explained the first time they are referenced in equations or text;

c) the free-space wave impedance is usually denoted as Z0. There is no need to complicate the commonly known notations;

d) please use a colon instead of a period if the equations are a continuation of a sentence (e.g., line 213). Furthermore, please properly start new paragraphs after explaining the equation symbols (e.g., line 216);

e) unit and value should be separated by a space (please use 214  instead of 214, etc.);

f) other editorial corrections are necessary according to the guidelines.

Author Response

Dear Reviewer 1, 

Please find our responses to your reviews as enclosed. Thank you.

Sincerely,

The authors

Reviewer 2 Report

Comments and Suggestions for Authors

This work present a holographic antenna design with good simulation results and experimental verification. However, the antenna design seems to be specific for a certain radiation pattern. How to design reconfigurable one to make the design more applicable? In addition, sidelobes in experimental results seems large, please explain this issue and provide possible solutions.

Comments on the Quality of English Language

There are some typos and grammar erros in the manuscript, for example: Line 14, have had had.

Author Response

Dear Reviewer 2, 

Please find our responses to your reviews as enclosed. Thank you.

Sincerely,

The authors

Round 2

Reviewer 1 Report

Comments and Suggestions for Authors

Thank you to the Authors for their answers and change. In my opinion, the manuscript can be accepted after minor editorial corrections:

a) the formatting of variables should be unified to a contemporary standard. This means, that all symbols used in text/equations/figures/tables should be formatted uniformly: italics for scalar variables, bold non-italic (already without the upper arrows) for matrices/vectors in the whole manuscript. In its current form, the mathematical notation is not very clear;

b) other editorial corrections are necessary according to the guidelines.

Author Response

Dear Academic Editor,

Please find our revised manuscript and a response letter that addresses the reviewers’ comments as enclosed. Thank you.

Sincerely,

The authors
